# Safety and continued use of the levonorgestrel intrauterine system as compared with the copper intrauterine device among women living with HIV in South Africa: A randomized controlled trial

**Catherine S. Todd**[1]*, **Heidi E. Jones**[2], **Nontokozo Langwenya**[3], **Donald R. Hoover**[4], **Pai-Lien Chen**[5], **Gregory Petro**[6,7], **Landon Myer**[3,8]

1 Maternal and Child Health and Nutrition Department, Global Health, Population, and Nutrition, Durham, North Carolina, United States of America, 2 Department of Epidemiology and Biostatistics, City University of New York School of Public Health, New York, New York, United States of America, 3 Division of Epidemiology and Biostatistics, School of Public Health and Family Medicine, University of Cape Town, Observatory, Cape Town, South Africa, 4 Department of Statistics and Institute for Health Care Policy and Aging Research, Rutgers University, Piscataway, New Jersey, United States of America, 5 Global Population and Health Research Department, Global Health, Population, and Nutrition, Durham, North Carolina, United States of America, 6 Department of Obstetrics and Gynaecology, University of Cape Town, Observatory, Cape Town, South Africa, 7 New Somerset Hospital, Cape Town, South Africa, 8 Division of Epidemiology and Biostatistics, Centre for Infectious Diseases Epidemiology and Research, School of Public Health and Family Medicine, University of Cape Town Observatory, Cape Town, South Africa

* ctodd@fhi360.org

## Abstract

### Background

Women living with HIV (WLHIV) have lower rates of contraceptive use than noninfected peers, yet concerns regarding contraceptive efficacy and interaction with antiretroviral therapy (ART) complicate counseling. Hormonal contraceptives may increase genital tract HIV viral load (gVL) and sexual transmission risk to male partners. We compared gVL, plasma VL (pVL), and intrauterine contraceptive (IUC) continuation between the levonorgestrel intrauterine system (LNG-IUS) and copper intrauterine device (C-IUD) in Cape Town, South Africa.

### Methods and findings

In this double-masked, randomized controlled noninferiority trial, eligible WLHIV were ages 18–40, not pregnant or desiring pregnancy within 30 months, screened and treated (as indicated) for reproductive tract infections (RTIs) within 1 month of enrollment, and virologically suppressed using ART or above treatment threshold at enrollment (non-ART). Between October 2013, and December 2016, we randomized consenting women within ART groups, using 1:1 permuted block randomization stratified by ART use, age (18–23, 24–31, 32–40), and recent injectable progestin contraceptive (IPC) exposure, and provided the allocated IUC. At all visits, participants provided specimens for gVL (primary outcome), pVL, RTI, and

**Data Availability Statement:** The de-identified database and variable specification files are available on DataVerse at the following link: https://dataverse.harvard.edu/dataset.xhtml?persistentId=doi:10.7910/DVN/NTN7KY

**Funding:** This study was funded by the Eunice Kennedy Shriver National Institute of Child Health and Development (R01HD071804; CST and LM) and by the US Agency for International Development Prevention Technologies agreement (No. GHO-A-00-09-00016-00; NA). We received in-kind donations from Bayer Pharmaceuticals (Mirena LNG-IUS), the Western Cape Government (C-IUDs), Sekisui Diagnostics (a portion of Osom BV Blue and Trichomonas), Cepheid Inc (a portion of Xpert CT/NG cartridges), and Alere-Abbott (Determine Syphilis RDTs). The funders had no role in study design, data collection and analysis, decision to publish, or preparation of the manuscript.

**Competing interests:** The authors have declared that no competing interests exist.

**Abbreviations:** AE, adverse event; aHR, adjusted hazard ratio; aOR, adjusted odds ratio; aRR, adjusted risk ratio; ART, antiretroviral therapy; AT, as-treated; BV, bacterial vaginosis; CI, confidence interval; C-IUD, copper T-380 intrauterine device; CT, *Chlamydia trachomatis*; DMPA, depot medroxyprogesterone acetate; GEE, generalized estimating equation; GLM, generalized linear model; gVL, genital tract viral load; IPC, injectable progestin contraceptive; ITT, intent-to-treat; IUC, intrauterine contraceptive; LARC, long-acting reversible contraceptive; LNG-IUS, levonorgestrel intrauterine system; LoD, limit of detection; LTFU, lost to follow-up; NA, not applicable; NG, *Neisseria gonorrhea*; OR, odds ratio; PID, pelvic inflammatory disease; pVL, plasma viral load; p-y, person-years; RDT, rapid diagnostic test; RTI, reproductive tract infection; SAE, serious adverse event; TB, tuberculosis; TP, *Treponema pallidum*; TV, *Trichomonas vaginalis*; WLHIV, women living with HIV.

pregnancy testing. We assessed gVL and pVL across 6 and 24 months controlling for enrollment measures, ART group, age, and RTI using generalized estimating equation and generalized linear models (non-ART group pVL and hemoglobin) in as-treated analyses. We measured IUC discontinuation rates with Kaplan-Meier estimates and Cox proportional hazards models. We enrolled 71 non-ART (36 LNG-IUS, 31 C-IUD; 2 declined and 2 were ineligible) and 134 ART-using (65 LNG-IUS, 67 C-IUD; 1 declined and 1 could not complete IUC insertion) women. Participant median age was 31 years, and 95% had 1 or more prior pregnancies. Proportions of women with detectable gVL were not significantly different comparing LNG-IUS to C-IUD across 6 (adjusted odds ratio [AOR]: 0.78, 95% confidence interval [CI] 0.44–1.38, $p = 0.39$) and 24 months (AOR: 1.03, 95% CI: 0.68–1.57, $p = 0.88$). Among ART users, proportions with detectable pVL were not significantly different at 6 (AOR = 0.83, 95% CI 0.37–1.86, $p = 0.65$) and 24 months (AOR = 0.94, 95% CI 0.49–1.81, $p = 0.85$), whereas among non-ART women, mean pVL was not significantly different at 6 months (−0.10 $\log_{10}$ copies/mL, 95% CI −0.29 to 0.10, $p = 0.50$) between LNG-IUS and C-IUD users. IUC continuation was 78% overall; C-IUD users experienced significantly higher expulsion (8% versus 1%, $p = 0.02$) and elective discontinuation (adjusted hazard ratio: 8.75, 95% CI 3.08–24.8, $p < 0.001$) rates. Sensitivity analysis adjusted for differential IUC discontinuation found similar gVL results. There were 39 serious adverse events (SAEs); SAEs believed to be directly related to IUC use ($n = 7$) comprised 3 pelvic inflammatory disease (PID) cases and 4 pregnancies with IUC in place with no discernible trend by IUC arm. Mean hemoglobin change was significantly higher among LNG-IUS users across 6 (0.57 g/dL, 95% CI 0.24–0.90; $p < 0.001$) and 24 months (0.71 g/dL, 95% CI 0.47–0.95; $p < 0.001$). Limitations included not achieving non-ART group sample size following change in ART treatment guidelines and truncated 24 months' outcome data, as 17 women were not yet eligible for their 24-month visit at study closure. Also, a change in VL assay during the study may have caused some discrepancy in VL values because of different limits of detection.

## Conclusions

In this study, we found that the LNG-IUS did not increase gVL or pVL and had low levels of contraceptive failure and associated PID compared with the C-IUD among WLHIV. LNG-IUS users were significantly more likely to continue IUC use and had higher hemoglobin levels over time. The LNG-IUS appears to be a safe contraceptive with regard to HIV disease and may be a highly acceptable option for WLHIV.

## Trial registration

ClinicalTrials.gov NCT01721798.

## Author summary

### Why was this study done?

- Women living with HIV (WLHIV) infection must consider their contraceptive options in tandem with and complicated by potential interactions between hormonal

contraception and HIV disease. These issues potentially contribute to lower use of highly effective hormonal contraceptive methods among WLHIV.

- Global use of intrauterine contraception (IUC), specifically the levonorgestrel intrauterine system (LNG-IUS), has increased across the last decade. However, there are few trials comparing safety of the LNG-IUS with that of the copper intrauterine device (C-IUD) among WLHIV and none focusing specifically on impact on HIV disease.

### What did the researchers do and find?

- In a randomized controlled trial in Cape Town, South Africa, 199 women with confirmed HIV seropositivity and not desiring pregnancy in the next 30 months were recruited and allocated, with 98 receiving the C-IUD and 101 the LNG-IUS.

- Participants had genital tract samples collected by menstrual cup and blood taken for genital and plasma viral load levels at enrollment and at 3-, 6-, 12-, 18-, and 24-month visits. Women could return at any time for unscheduled visits and remained in the study even if the IUC was removed on request.

- Women receiving the LNG-IUS did not have any significant difference in change in detectable genital tract viral load compared with C-IUD users, regardless of ART status. However, women receiving the C-IUD were more likely to have the IUC removed than those receiving the LNG-IUS through the study period.

### What do these findings mean?

- This study is among the first comparing a hormonal to nonhormonal long-acting reversible contraceptive method with genital tract HIV RNA as the primary outcome among WLHIV.

- These data suggest that LNG-IUS is as safe as the C-IUD for WLHIV and will strengthen international medical eligibility guidelines.

- The high continuation rates of the LNG-IUS, critically important for a device with a 5-year duration of use, should prompt including this method in the available method mix to promote the ability of WLHIV to meet their fertility goals.

## Introduction

Across sub-Saharan Africa, many women acquire HIV early in their reproductive years. Subsequently, women living with HIV (WLHIV) must balance contraceptive decision-making with the potential impact of HIV infection on pregnancy outcomes and timing, risk of transmission to partners or infants, and appropriate method choice for prevention of pregnancy and sexually transmitted infections [1,2]. WLHIV discontinue hormonal methods more frequently and may have higher rates of unmet contraceptive need than their uninfected peers [3–5].

Intrauterine contraceptives (IUCs) have some of the longest effective durations among long-acting reversible contraceptive (LARC) methods but are not widely used in countries with both high total fertility rates and high HIV prevalence [6,7]. The copper T-380

intrauterine device (C-IUD) is a highly effective contraceptive method that appears to be safe for WLHIV; a Kenyan study among 156 WLHIV not using antiretroviral therapy (ART) found a low incidence of ascending reproductive tract infection (RTI) and no measurable impact on HIV progression [8,9]. However, less is known about levonorgestrel intrauterine system (LNG-IUS) safety for WLHIV.

Systematic reviews and analysis from a large observational cohort suggest that some hormonal contraceptives may be associated with increased HIV progression among or transmission to sexual partners of WLHIV [10,11]. Genital tract viral load (gVL) has been implicated in HIV transmission to male partners and a host of factors are noted to impact gVL, potentially including exogenous progestins [12–14]. LNG-IUS users experience sustained low serum LNG concentrations and relatively higher local LNG tissue concentrations [15,16], but few studies have evaluated genital tract HIV RNA shedding among WLHIV using the LNG-IUS. One small case series evaluated 12 WLHIV, 10 of whom were receiving ART in Finland for 12 months post-LNG-IUS insertion [17], and another followed 25 WLHIV not using ART in Kenya for 6 months [18]. In these studies, women served as their own controls, and the presence or quantity of plasma and genital tract HIV RNA was compared over time, with no significant change noted in either study. Similarly, for the C-IUD, Richardson and colleagues compared samples from 96 WLHIV not using ART immediately prior to and 6 months after C-IUD placement and found no significant change in cervical HIV RNA detection [19]. Although these studies are notable for relatively high IUC continuation rates, most were of short duration, had small sample sizes, and lacked a comparator group, relying on a crossover design. A systematic review including these and other smaller studies concluded that genital tract shedding likely does not increase with IUC use among WLHIV but that supporting evidence quality regarding shedding is fair to poor [20].

Data for whether IUCs are safe for use among WLHIV and, specifically, whether IUCs impact gVL is germane given expanded global ART use and findings of persistent detectable gVL among WLHIV virally suppressed with very low to undetectable plasma viral load (pVL) [21–23]. There are emerging concerns that HIV acquisition and transmission risk may increase in the presence of RTIs, non-*Lactobacillus* dominant microbiota, and hormonal contraceptives because of local interaction with ART agents [24–26]. Thus, evidence regarding gVL change with exposure to hormonal and nonhormonal IUC use among WLHIV with typical ART use patterns in a context with high background RTI prevalence is timely and may help inform this debate. The purpose of this study was to compare the LNG-IUS with the C-IUD among WLHIV to assess safety concerning HIV transmission and disease progression using the proxy measures of gVL (primary outcome) and pVL (secondary outcome) and by adverse events (AEs) and to assess acceptability as measured by IUC continuation (secondary outcomes). Our hypothesis was that the LNG-IUS was no different from C-IUD regarding change in rates of detectable gVL (noninferiority trial).

## Methods

We conducted a double-masked randomized controlled trial with one-to-one allocation comparing the C-IUD (SMB Copper T-380A IUD; SMB Corporation of India, Mumbai, India, or Nova-T Copper T-380 IUD; Bayer Pharmaceuticals, Germany) to the LNG-IUS (Mirena; Bayer Health Care Pharmaceuticals, Montville, NJ) among WLHIV, stratified by enrollment ART status (ART and non-ART users) in Gugulethu, Cape Town, South Africa (clinicaltrials. gov NCT01721798) (please see S1 CONSORT checklist). Women were recruited from surrounding healthcare facilities, community events promoting reproductive health, and radio and newspaper advertisements.

## Ethical review

The University of Cape Town Human Research Ethics Committee (#283/2012) and FHI 360 Protection of Human Subjects Committee (#10369/398733) approved the study prior to study activity initiation, and all participants provided written informed consent prior to screening and trial participation.

## Eligibility

Eligible women were ages 18–40 years; confirmed as HIV-infected; not pregnant or desiring pregnancy for the next 30 months; at least 6 months postpartum; not planning relocation within 30 months; screened with cervical cytology within the last year (women without a documented cervical cytology report within the last year were offered cytology smear during the pelvic examination at the screening visit in an opt-out fashion to ensure appropriate triage for evidence of neoplasia and eligibility at enrollment visit); without history of ectopic pregnancy, tubal sterilization, or other conditions contraindicating IUC use; and interested in IUC use for contraception. Eligible women using ART had documented pVL < 1,000 copies/mL in the last 6 months, and non-ART women were ART-ineligible at enrollment by CD4 lymphocyte count per local guidelines. As guidelines changed during the trial period, the non-ART arm criteria also changed; when guidelines changed to immediate ART initiation, non-ART women were screened and enrolled and referred for ART at these visits.

## Visit procedures

The visit schedule comprised screening, enrollment, and follow-up at 3, 6, 12, 18, and 24 months. All visits were conducted at the study clinic on site at the Gugulethu Community Health Centre in Gugulethu, Cape Town, South Africa. At screening, consenting women were tested for RTIs (*Neisseria gonorrhea* [NG], *Chlamydia trachomatis* [CT], *Trichomonas vaginalis* [TV], *Treponema pallidum* [TP], and sialidase-positive bacterial vaginosis [BV]) and treated for reactive test results. At enrollment, women had either nonreactive results for all pathogens within 30 days or were 2 to 4 weeks posttreatment for reactive tests; women could be screened up to 3 times. Women were encouraged to continue their current contraceptive method until enrollment; for those reporting injectable progestin contraceptive (IPC) use (depot medroxyprogesterone acetate [DMPA] or norethisterone enanthate), enrollment was scheduled synchronously with the next IPC dose to mitigate potential residual hormonal effect on gVL measures while sustaining contraception [10]. Staff scheduled visits to avoid menstrual bleeding, with visits rescheduled for reported or clinically observed bleeding, and asked participants to avoid vaginal intercourse or douching 48 hours prior to visits.

At enrollment, participants completed an interview-administered baseline questionnaire on sociodemographic characteristics, reproductive and sexual history, menstrual bleeding, and pelvic symptoms; follow-up visit questionnaires also queried IUC acceptability and changes in partnership status and HIV care. At all visits, women inserted menstrual cups (Instead Softcup; Evofem, San Diego, CA, United States of America) for genital tract sample collection, provided urine specimens for pregnancy testing, and provided blood for pVL. At all visits except 3 months, full blood count and CD4 testing (non-ART group) were also conducted. A study nurse certified in IUC insertion provided the LNG-IUS or C-IUD at the end of the enrollment visit based on randomized allocation. Each visit included pelvic examination to collect swabs for RTI testing, bimanual exam, and, at follow-up visits, string visualization to confirm IUC placement. If strings were not visible, immediate onsite ultrasound was used to confirm IUC presence and intrauterine placement. During visits, the study nurse inquired regarding bleeding or other symptoms, which were tracked within an AE register.

Participants were counseled about possible bleeding pattern changes, how to check strings, and notifying the clinic immediately if the IUC was expelled, heavy or abnormal bleeding occurred, or symptoms concerning for RTI or pregnancy occurred. Women were further counseled they could have the IUC removed at any time and a new method provided. Women requesting IUC removal and agreeing to continue with the study were retained through the 24-month visit but excluded from all as-treated (AT) analyses; the only participants terminated from the study were those with diagnosed pregnancy. Women presenting within 72 hours of complete IUC expulsion or with recognized partial expulsion on examination were offered replacement with the allocated IUC if a pregnancy test was negative. Women with clinically diagnosed pelvic inflammatory disease (PID) were offered IUC removal and treated according to national protocols [27]; these women received close follow-up.

## Laboratory testing

Plasma and genital tract specimens were tested for HIV RNA at the South African National Health Laboratory Service (NHLS) using the Abbott M2000SP/RT VL assay (Abbott Diagnostics, IL, USA) until June 2015, then subsequently with Roche COBAS TaqMan HIV-1 v2·0 assay (Roche Diagnostics, Mannheim, Germany) with lower limits of detection (LoDs) of 40 and 20 copies/mL, respectively [28]. RTI testing comprised NG and CT nucleic acid amplification testing with NG/CT Xpert (Cepheid Diagnostics, Sunnyvale, CA, USA), TV and BV rapid diagnostic tests (RDTs) for genital tract specimens with OSOM Trichomonas and BV Blue (Sekisui Diagnostics, Lexington, MA, USA), and TP with Alere Determine Syphilis (Alere Diagnostics, San Diego, CA, USA) RDT for whole blood with reflex Rapid Plasma Reagin titer testing of plasma, according to package instructions. Hematological testing comprised CD4 lymphocyte count (non-ART group) using Beckman Coulter XL flow cytometry and complete blood count tests, including hemoglobin, all processed on automated platforms per NHLS procedures.

## Outcome measures

The primary trial outcome was change in proportions of WLHIV with detectable gVL from baseline across 6 and 24 months of follow-up. At study initiation in 2013, the original primary outcome was change in pVL, and enrollment into the study was originally restricted to non-ART users. With changes in South African guidelines to ART initiation at diagnosis in 2014 [29], we amended the primary outcome to genital HIV shedding, as a proxy for transmission (originally a secondary outcome), and enrollment to include virally suppressed (pVL<1,000 copies/mL) ART-using WLHIV. Secondary outcomes were changes in pVL; other side effects and safety issues related to IUC use, including hemoglobin, overall and related AEs, and serious AEs (SAEs); and acceptability via IUC continuation.

## Sample size rationale and adjustment with outcome change

The original sample size and assumptions are presented in S1 Text. With the change in primary outcome in 2014, the original evaluable sample size of 288 was divided into ART and non-ART groups, and power estimates were calculated for change in proportion of WLHIV with detectable gVL between study arms using a 2-sample exact test with enrollment and 6-month visit measures, using a 2-sided α = 0.05. We further assumed a 15% IUC discontinuation rate and a 5% loss to follow-up rate at 6 months. For ART-using women, we estimated that 54 evaluable women in each arm gave 80% power to exclude a difference of at least 27.7% in proportion of women with detectable gVL between arms at 6 months, assuming 22% of ART-using women had detectable gVL [30]. For non-ART women, we estimated that 62

evaluable women in each arm would provide 80% power to measure a 23.7% difference in detectable gVL between arms at 6 months, informed by shedding rates of 61% of the first 36 non-ART participants [22]. For the pooled sample, we estimated 80% power to detect at least a 19.1% difference in detectable gVL with 116 evaluable women in each arm, assuming a pooled baseline gVL of 42.8%. The pooled gVL value was calculated by using non-ART gVL of 61% * 62/116 + estimated ART-user gVL of 22% * 54/116; this was calculated in 2015 at the time of primary outcome change and ART group addition with available evidence [22].

## Randomization and masking

We used 1:1 permuted block randomization with block sizes of 4 to 6 stratified by ART use, age (18–23, 24–31, 32–40), and recent IPC exposure. An external data manager generated the randomization scheme and provided the study site with sequentially numbered, sealed envelopes by stratification arm. The study manager or nurse provided each eligible participant the next sequential envelope in their stratum during the enrollment visit, specifying either "Treatment A" or "Treatment B." Only study nurses conducting insertions and a site manager were unmasked to specific IUC corresponding to arm; all investigators and outcomes assessors were masked. Participants were unmasked at exit visits.

## Statistical analysis

Participant baseline demographic and health history data by study arm were summarized using means (± standard deviations), medians (with interquartile ranges), and proportions. IUC expulsions presenting beyond 72 hours or women declining IUC replacement were censored from VL-associated analysis at that time point for AT analyses. Women with positive pregnancy tests had immediate ultrasound and referral as appropriate with IUC removal for cases with partial expulsion; these women were censored from all analyses following pregnancy diagnosis.

For primary and secondary HIV viral load outcomes, we analyzed differences between arms both by AT and intent-to-treat (ITT) using generalized linear models (GLMs) with generalized estimating equations (GEEs) via combined and stratified analysis by ART group. We preferentially selected AT analyses for the primary (gVL) and 2 secondary outcomes (pVL and AEs); ITT analyses are also presented. We selected AT analysis during study design and analysis planning, as the VL outcomes, pVL and gVL (the latter becoming the primary outcome measure), are both safety measures reflective of potential accelerated disease progression and transmission to partners, respectively. As safety outcomes, we believed it was essential to reflect actual exposure to IUCs with the AT approach rather than the more conservative ITT. To compare the primary outcome, change in proportion of women with detectable gVL from baseline pooled across 6 and 24 months, we used GEE with logistic link and exchangeable working correlation structure to estimate crude and adjusted odds ratios (ORs) with 95% confidence intervals (CIs). All models were adjusted for baseline gVL, as the primary outcome was change in gVL, post-IUC insertion, and adjusted for age and ART group, given stratified randomization. In sensitivity analysis, we further adjusted AT models for prespecified hypothesized confounders of concurrent RTI and baseline pVL. Because of differential IUC discontinuation rates between arms, sensitivity analyses using inverse probability-of-discontinuation weights were applied to primary and secondary HIV-related safety outcomes [31], as were sensitivity analyses testing visit month effect and independent working correlation structure.

For analysis of changes in pVL, 2 approaches were used. For ART users, we used GEE to generate ORs with 95% CIs to evaluate detectable pVL rates between arms from baseline

pooled across 6 and 24 months. For non-ART women, we used GLM with identical link to estimate mean change in $\log_{10}$ pVL between arms from baseline pooled across 6 months; we did not analyze across 24 months, as many non-ART women initiated ART during that period. The adjusted models for both ART groups included the previously prespecified confounders and baseline VL measures. For analyses of VL as a continuous measure with $\log_{10}$ values, participants with "undetectable" or lower than LoD results were assigned the value of 20 copies/mL, representing the lowest LoD of the 2 commercial assays used for VL measures and the midpoint between 0 and LoD of the assay with the higher LoD. Previous studies have used either approach (the manufacturer's LoD to quantify undetectable measures [18,32–34] or the value between 0 and the LoD for undetectable values [35–37]). As VL tests were run through NHLS and assays were changed midway through the study, we believe using the value of 20 copies/mL represents the best assigned value for undetectable VL measures.

For the secondary acceptability outcome of discontinuation, we used Kaplan-Meier models with log-rank tests to estimate and compare cumulative discontinuation probabilities between arms, both overall and stratified by ART group. We also estimated crude and adjusted hazard ratios (aHRs) of discontinuation, adjusting for prespecified covariates of age, ART status, and concurrent RTI, using Cox proportional hazards models. The proportional hazard assumptions of Cox models were examined using Schoenfeld residuals, which do not reject that assumption with $p = 0.134$. AEs were summarized by those considered related to IUC use and by system organ class, pooled across study arms. Percentages of participants experiencing each AE (1 or more times) were calculated, and we analyzed differences between arms using chi-squared or, for comparisons with 5 or fewer individuals in 1 group, Fisher's exact tests. We describe SAEs deemed related to IUC use by arm and with rates based on time contributed in an AT model. We compared mean absolute change in hemoglobin (g/dL) between arms across 6 and 24 months in pooled and ART group-stratified analysis using the same approach as for non-ART group pVL analysis. Statistical analyses were performed using SAS 9.4 (SAS Institute, Cary, NC). For further details on analytic approach and rationale, please see S1 Text for the study protocol and S2 Text for the statistical analysis plan.

## Results

### Participant characteristics and baseline measures

Of 205 women enrolled from October 2013 to December 2016, 199 contributed data for analysis through study closure in July 2018 (Fig 1).

More than 85% of participants continued through the 6-month and end-of-study visits. Because of clinical activity closure in July 2018, 17 women (16 ART users, 1 non-ART user) were not eligible for the 24-month visit; as such, we did not conduct a 24-month visit with VL and RTI measures for these participants. Instead, to ensure informed contraceptive decision-making, we conducted exit visits to unmask these participants and ask whether they wished to retain their IUC. We also did telephone follow-up with these participants at the time of 24-month visit eligibility to assess IUC retention in women who elected to continue their allocated IUC at exit visit.

Most participants had been pregnant at least once and had a mean age of 31.4 years (Table 1). Only one-fifth had completed secondary education, and more than 80% reported having 1 partner over the last 12 months. At screening, RTIs were common, diagnosed among approximately 40% of women, particularly in the non-ART group, with BV, TV, and CT being the most frequent RTIs detected. At enrollment, more than 10% of ART users and 55% of non-ART women had detectable gVL; there were no significant differences in proportion of women with detectable gVL between study arms within each ART group.

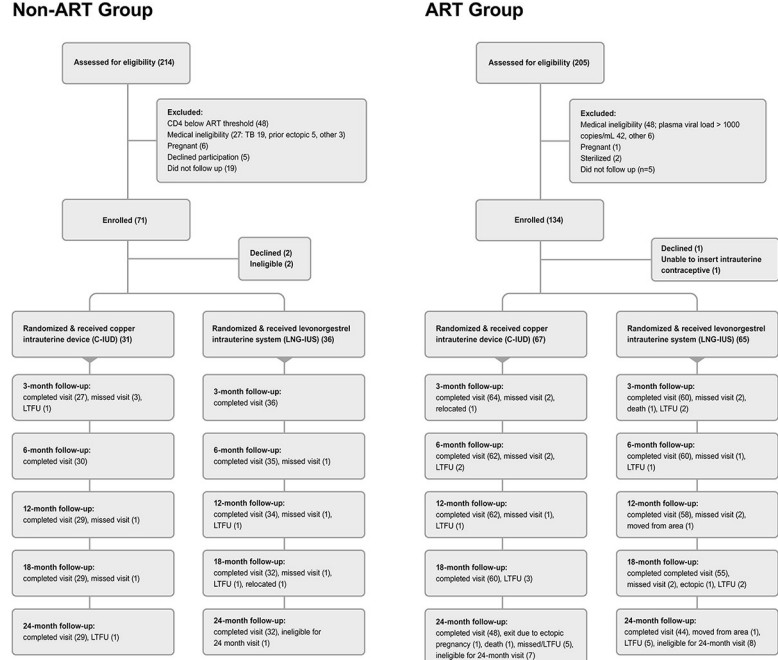

**Fig 1. Trial recruitment and study activity flow chart.** ART, antiretroviral therapy; C-IUD, copper intrauterine device; LNG-IUS, levonorgestrel intrauterine system; LTFU, loss to follow-up; TB, tuberculosis.

## Safety regarding genital tract and plasma HIV RNA

Detectable levels of genital tract HIV RNA were consistently present in at least 8% of participants at any time point, irrespective of ART status (Fig 2).

Approximately one-fifth of ART users had detectable pVL at enrollment, and this increased steadily within this group across the cohort period, mirrored by detectable gVL. Larger proportions of women in the non-ART group at enrollment had undetectable or suppressed pVL values starting from 18 months' follow-up, with related trends in gVL.

There were no significant differences between study arms in proportions of women with detectable gVL in all models and by ART group (Table 2). In sensitivity analyses including censoring adjustment for differential IUC discontinuation, adjustment for visit month effect, and independent working correlation, we found no substantive change in odds of detectable gVL between arms (S1–S3 Tables).

We also detected no significant difference in detectable gVL between baseline compared with combined 3- and 6-month visit measures for each IUC (Table 3).

For the secondary pVL outcome, there was no significant difference in odds of detectable pVL between arms for any analytic approach among ART users (Table 4). For non-ART users, there was no significant difference in mean change in $\log_{10}$ pVL across 6 months; we did not analyze end visit data, as 18.0% ($n = 11$) of non-ART women reported initiating ART by the exit visit, with 6.3% of initiations occurring by the 12-month visit. These findings were similar in sensitivity analyses identical to those used for gVL (S4–S6 Tables).

## Safety regarding reproductive and systemic health measures

At least 1 AE was reported by 98% of participants in each arm. Specific AEs potentially related to IUC use were common, and some differed significantly between study arms, with

**Table 1. Participant baseline sociodemographic and health characteristics at enrollment by arm and ART status ($n$ = 199).**

| | All participants ($n$ = 199) | | ART-using women ($n$ = 132) | | Non-ART women ($n$ = 67) | |
|---|---|---|---|---|---|---|
| **Variable** | **C-IUD** ($n$ = 98) | **LNG-IUS** ($n$ = 101) | **C-IUD** ($n$ = 67) | **LNG-IUS** ($n$ = 65) | **C-IUD** ($n$ = 31) | **LNG-IUS** ($n$ = 36) |
| Age in years, mean (SD) | 31.4 (4.6) | 31.4 (4.9) | 31.8 (4.5) | 32.1 (4.8) | 30.6 (4.6) | 30·2 (5.0) |
| Completed secondary education, $n$ (%) | 21 (21.4) | 28 (27.7) | 16 (23.9) | 15 (23.1) | 5 (16.1) | 13 (36.1) |
| Currently employed, $n$ (%) | 29 (29.6) | 38 (37.6) | 18 (26.9) | 22 (33.8) | 11 (35.5) | 16 (44.4) |
| Ever pregnant, $n$ (%) | 96 (98.0) | 94 (93.1) | 65 (97.0) | 63 (96.9) | 31 (100.0) | 31 (86.1) |
| Number of pregnancies, median (IQR) | 2 (1, 3) | 2 (1, 3) | 2 (1, 3) | 2 (2, 3) | 2 (1, 3) | 2 (1, 2) |
| Number of sex partners in past 12 months, $n$ (%) | | | | | | |
| 0 | 6 (6.1) | 4 (4.0) | 5 (7.5) | 3 (4.6) | 1 (3.2) | 1 (2.8) |
| 1 | 82 (83.7) | 87 (86.1) | 55 (82.1) | 57 (87.7) | 27 (87.1) | 30 (83.3) |
| 2+ | 10 (10.2) | 10 (9.9) | 7 (10.4) | 5 (7.7) | 3 (9.7) | 5 (13.9) |
| Sexual frequency, last 3 months, $n$ (%) | | | | | | |
| Less than once per month | 25 (25.8) | 29 (28.7) | 14 (21.2) | 18 (27.7) | 11 (35.5) | 11 (30.6) |
| 1–3 times per month | 24 (24.7) | 22 (21.8) | 18 (27.3) | 12 (18.5) | 6 (19.4) | 10 (27.8) |
| Once per week | 16 (16.5) | 24 (23.8) | 9 (13.6) | 18 (27.7) | 7 (22.6) | 6 (16.7) |
| More than once per week | 32 (33.0) | 26 (25.7) | 25 (37.9) | 17 (26.1) | 7 (22.6) | 9 (25.0) |
| Years since HIV diagnosis, median (IQR) | 5.4 (2.0, 8.4) | 5.0 (2.0, 8.4) | 7.3 (3.3, 10.4) | 6.2 (4.2, 9.0) | 2.6 (1.1, 4.3) | 3.8 (0.5, 5.9) |
| Detectable pVL, $n$ (%) | 44 (44.9) | 46 (45.5) | 14 (20.9) | 12 (18.5) | 30 (96.8) | 34 (94.4) |
| pVL copies/mL, $n$ (%) | | | | | | |
| <40 | 55 (56.1) | 55 (54.5) | 54 (80.6) | 53 (81.5) | 1 (3.2) | 2 (5.6) |
| 40–1,000 | 11 (11.2) | 13 (12.9) | 6 (9.0) | 8 (12.3) | 5 (16.1) | 5 (13.9) |
| 1,001–10,000 | 15 (15.3) | 17 (16.8) | 5 (7.5) | 2 (3.1) | 10 (32.3) | 15 (41.7) |
| 10,001–100,000 | 15 (15.3) | 13 (12.9) | 2 (3.0) | 2 (3.1) | 13 (41.9) | 11 (30.6) |
| >100,000 | 2 (2.0) | 3 (3.0) | 0 (0.0) | 0 (0.0) | 2 (6.4) | 3 (8.3) |
| **pVL $\log_{10}$ copies/mL, median (IQR)[a]** | 3.7 (2.8, 4.5) | 3.5 (2.9, 4.2) | 3.2 (2.2, 3.6) | 2.2 (1.9, 3.1) | 4·0 (3.3, 4.5) | 3.8 (3.1, 4.3) |
| CD4 lymphocyte count at last screening prior to enrollment, median (IQR)[b] | NA | NA | NA | NA | 684 (551, 832) | 568 (462, 732) |
| Detectable gVL, $n$ (%) | 28 (28.6) | 30 (29.7) | 10 (14.9) | 7 (10.8) | 18 (58.1) | 23 (63.9) |
| **gVL copies/mL, $n$ (%)[c]** | | | | | | |
| <40 | 71 (72.4) | 71 (70.3) | 57 (85.1) | 58 (89.2) | 14 (45.2) | 13 (36.1) |
| 40–1,000 | 8 (8.2) | 13 (12.9) | 7 (10.4) | 5 (7.7) | 1 (3.2) | 8 (22.2) |
| 1,001–10,000 | 12 (12.2) | 9 (8.9) | 2 (3.0) | 1 (1.5) | 10 (32.3) | 8 (22.2) |
| 10,001–100,000 | 7 (7.1) | 7 (6.9) | 1 (1.5) | 1 (1.5) | 6 (19.3) | 6 (16.7) |
| >100,000 | 0 (0.0) | 1 (1.0) | 0 (0.0) | 0 (0.0) | 0 (0.0) | 1 (2.8) |
| **gVL $\log_{10}$ copies/mL, median (IQR)[a]** | 3.2 (2.8, 4.0) | 3.4 (2.5, 4.0) | 2.6 (2.3, 3.3) | 2.5 (2.1, 3.6) | 3.3 (3.1, 4.1) | 3.6 (2.7, 4.2) |
| Reproductive tract infections at screening,[d] $n$ (%) | 30 (30.6) | 40 (39.6) | 17 (25.4) | 22 (33.8) | 13 (41.9) | 18 (50.0) |
| Sialidase-positive bacterial vaginosis | 11 (11.2) | 18 (17.8) | 5 (7.5) | 9 (13.8) | 6 (19.3) | 9 (25.0) |
| *N. gonorrhea* | 4 (4.1) | 4 (4.0) | 3 (4.5) | 2 (3.1) | 1 (3.2) | 2 (5.6) |
| *C. trachomatis* | 5 (5.1) | 7 (6.9) | 2 (3.0) | 5 (7.7) | 3 (9.7) | 2 (5.6) |
| *T. vaginalis* | 10 (10.2) | 9 (8.9) | 7 (10.4) | 6 (9.2) | 3 (9.7) | 3 (8.3) |
| *T. pallidum* | 0 (0.0) | 2 (2.0) | 0 (0.0) | 0 (0.0) | 0 (0.0) | 2 (5.6) |

(*Continued*)

**Table 1.** (Continued)

| Variable | All participants (*n* = 199) | | ART-using women (*n* = 132) | | Non-ART women (*n* = 67) | |
|---|---|---|---|---|---|---|
| | C-IUD (*n* = 98) | LNG-IUS (*n* = 101) | C-IUD (*n* = 67) | LNG-IUS (*n* = 65) | C-IUD (*n* = 31) | LNG-IUS (*n* = 36) |
| Hemoglobin, mean (SD)[e] | 12.4 (1.2) | 12.4 (1.1) | 12.2 (1.3) | 12.4 (1.1) | 12.8 (1.0) | 12.4 (1.3) |

[a]Among women with detectable (>40 copies/mL) pVL or gVL.

[b]Data not collected for ART-using women.

[c]Three cases imputed.

[d]Enrollment within 1 month of screening.

[e]One case missing data in ART-using C-IUD arm.

ART, antiretroviral therapy; C-IUD, copper intrauterine device; gVL, genital viral load; IQR, interquartile range; LNG-IUS, levonorgestrel intrauterine system; NA, not available; pVL, plasma viral load; SD, standard deviation

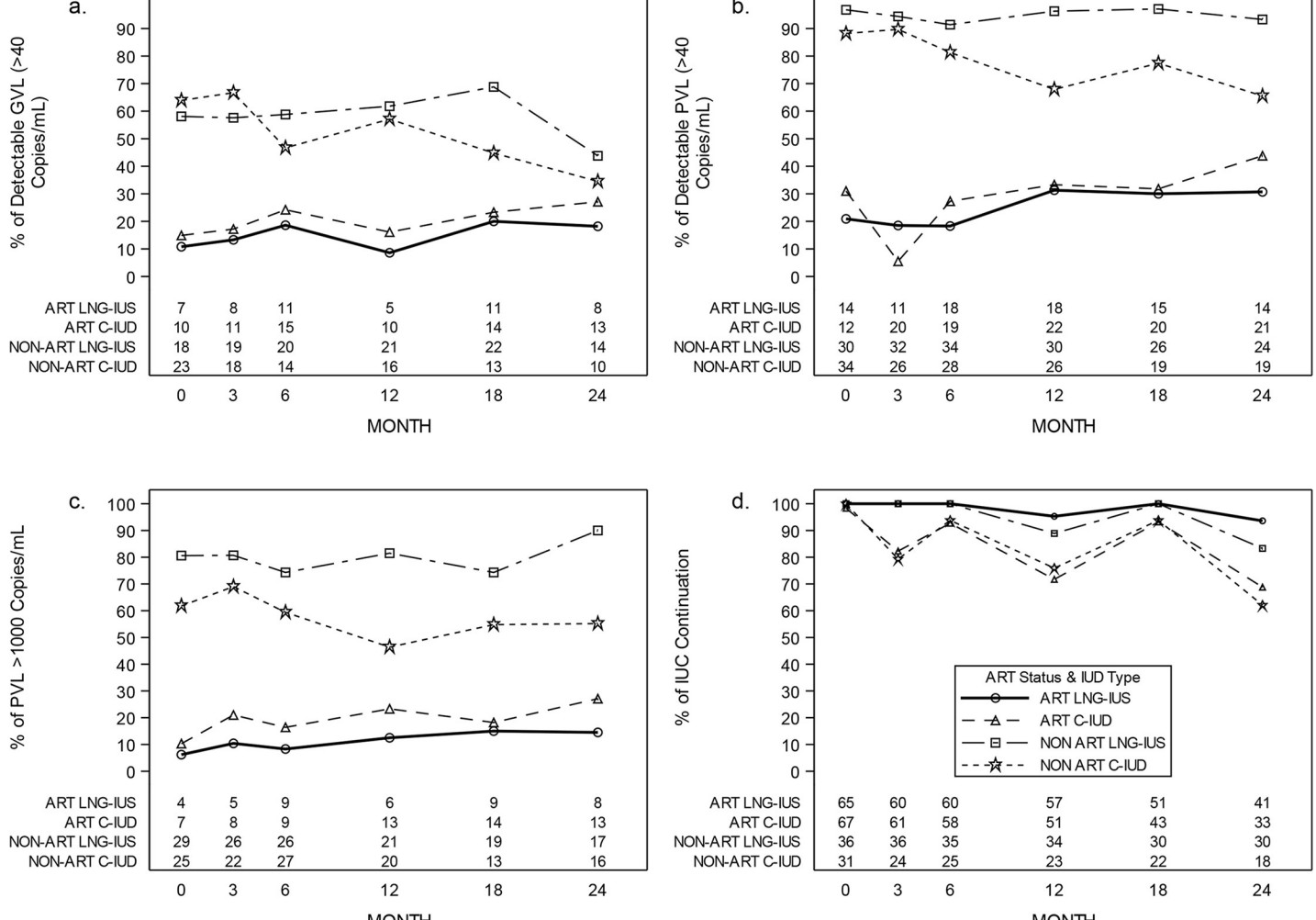

**Fig 2. Trends in female gVL and pVL and IUC continuation by study arm and within ART use strata across the cohort period overall and with AT group.** ART, antiretroviral therapy; AT, as-treated; C-IUD, copper intrauterine device; gVL, genital viral load; IUC, intrauterine contraceptive; LNG-IUS, levonorgestrel intrauterine system; pVL, plasma viral load.

**Table 2. Detectable genital tract HIV RNA rates across 6 and 24 months by ART status.**

| | All participants (n = 199) | ART-using women (n = 132) | Non-ART women (n = 67) |
|---|---|---|---|
| Detectable gVL by study visit | AOR (95% CI) *p*-value | AOR (95% CI) *p*-value | AOR (95% CI) *p*-value |
| **As-treated analysis[a]** | | | |
| Across 6 months | 0.82 (0.46–1.47) 0.51 | 0.85 (0.39–1.84) 0.68 | 0.79 (0.33–1.90) 0.60 |
| Across 24 months | 0.96 (0.63–1.45) 0.84 | 0.79 (0.46–1.36) 0.40 | 1.20 (0.61–2.35) 0.60 |
| **Intent-to-treat analysis[a]** | | | |
| Across 6 months | 0.87 (0.50–1.52) 0.62 | 0.76 (0.36–1.60) 0.47 | 1.04 (0.43–2.51) 0.94 |
| Across 24 months | 0.99 (0.66–1.48) 0.97 | 0.73 (0.44–1.24) 0.25 | 1.41 (0.74–2.67) 0.30 |
| **Adjusted as-treated analysis** | | | |
| Across 6 months[b] | 0.78 (0.44–1.38) 0.39 | 0.82 (0.38–1.76) 0.61 | 0.70 (0.30–1.68) 0.43 |
| Across 24 months[b] | 0.95 (0.63–1.44) 0.81 | 0.79 (0.46–1.35) 0.39 | 1.19 (0.61–2.33) 0.61 |
| Across 24 months[c] | 1.03 (0.68–1.57) 0.88 | 0.80 (0.48–1.33) 0.40 | 1.31 (0.61–2.84) 0.49 |

[a]Adjusted for baseline detectable gVL, age, and ART group (combined only).

[b]Adjusted for baseline detectable gVL, any RTI, age, and ART group (combined only).

[c]Adjusted for baseline detectable gVL, any RTI, age, baseline pVL (dichotomous), pVL ($log_{10}$ continuous), and ART group (combined only).

AOR, adjusted odds ratio; ART, antiretroviral therapy; CI, confidence interval; gVL, genital viral load; pVL, plasma viral load; RTI, reproductive tract infection

menorrhagia, dysmenorrhea, and anemia more prevalent among C-IUD users and amenorrhea more common among LNG-IUS users (Table 5). Non-ART women reported pelvic pain, dysmenorrhea, and bloating in higher proportions than ART users. There were 39 SAEs, including 2 deaths unrelated to IUC use (LNG-IUS: smoke inhalation; C-IUD: brief illness disclosed by family), 3 cases of PID (2 LNG-IUS, rate = 1.2/100 person-years [p-y], 95% CI 0.3–4.6), 1 C-IUD (rate = 0.7/100 p-y, 95% CI 0.1–4.9, *p* = 0.68; overall rate = 0.9/100 p-y, 95% CI 0.3–2.9), and 4 pregnancies with IUC in place (2 ectopic pregnancies in C-IUD users; 1 incomplete abortion with no IUC recovered in a C-IUD user, suggesting unrecognized expulsion; and 1 intrauterine pregnancy in an LNG-IUS user). Pregnancy rates with IUC in place were 1.3/100 p-y overall (95% CI 0.4–3.3), with 2.1/100 p-y (95% CI 0.7–6.5) for C-IUD and 0.6/100 p-y (95% CI 0.1–4.6) for LNG-IUS (*p* = 0.26). There were no recognized cases of uterine perforation.

**Table 3. Change in detectable genital tract HIV RNA viral load between baseline and pooled 3 and 6 months post-intrauterine contraceptive insertion measures, by ART status (n = 199).**

| | LNG-IUS | | | C-IUD | | |
|---|---|---|---|---|---|---|
| | Pooled | ART | Non-ART | Pooled | ART | Non-ART |
| | OR (95% CI) *p*-value | OR (95% CI) *p*-value | OR (95% CI) *p*-value | OR (95% CI) *p*-value | OR (95% CI) *p*-value | OR (95% CI) *p*-value |
| **As-treated analysis** | (n = 98) | (n = 62) | (n = 36) | (n = 88) | (n = 63) | (n = 25) |
| Crude | 1.02 (0.65–1.58) 0.94 | 1.49 (0.58–3.81) 0.41 | 0.76 (0.36–1.61) 0.48 | 1.17 (0.80–1.70) 0.43 | 1.30 (0.67–2.52) 0.44 | 1.02 (0.55–1.90) 0.94 |
| Adjusted | 0.87 (0.47–1.62) 0.66 | 1.18 (0.39–3.55) 0.77 | 0.66 (0.16–2.70) 0.57 | 1.05 (0.67–1.63) 0.84 | 1.19 (0.59–2.39) 0.63 | 0.81 (0.28–2.37) 0.70 |
| **Intent-to-treat analysis** | (n = 101) | (n = 65) | (n = 36) | (n = 98) | (n = 67) | (n = 31) |
| Crude | 1.04 (0.67–1.62) 0.86 | 1.56 (0.61–4.00) 0.36 | 0.76 (0.36–1.61) 0.47 | 1.16 (0.81–1.65) 0.43 | 1.20 (0.66–2.17) 0.55 | 1.22 (0.69–2.17) 0.49 |
| Adjusted | 0.94 (0.51–1.74) 0.85 | 1.40 (0.46–4.22) 0.55 | 0.66 (0.16–2.70) 0.57 | 1.08 (0.71–1.64) 0.71 | 1.08 (0.58–2.03) 0.81 | 1.17 (0.51–2.66) 0.71 |

[a]Adjusted for detectable pVL for ART group, pVL ($log_{10}$ continuous) for non-ART group.

ART, antiretroviral therapy; CI, confidence interval; C-IUD, copper intrauterine device; LNG-IUS, levonorgestrel intrauterine system; OR, odds ratio; pVL, plasma viral load

**Table 4. Plasma HIV RNA viral load outcomes by ART status across 6 and 24 months among women living with HIV using the LNG-IUS or C-IUD, stratified by ART use ($n$ = 199).**

| | Detectable pVL | Change of $\log_{10}$ pVL through 6-month visit |
|---|---|---|
| | ART group ($n$ = 132) | Non-ART group ($n$ = 67) |
| | AOR (95% CI), *p*-value | Difference (95% CI), *p*-value |
| **As-treated analysis** | | |
| Across 6 months[a] | 0.82 (0.36–1.83), 0.62 | |
| Across 6 months[b] | | −0.05 (−0.25 to 0.15), 0.67 |
| Across 24 months[a] | 0.93 (0.48–1.79), 0.82 | |
| **Intent-to-treat analysis** | | |
| Across 6 months[a] | 0.83 (0.37–1.86), 0.65 | |
| Across 6 months[b] | | −0.07 (−0.26 to 0.12), 0.59 |
| Across 24 months[a] | 0.90 (0.47–1.73), 0.76 | |
| **Adjusted as-treated analysis** | | |
| Across 6 months[c] | 0.83 (0.37–1.86), 0.64 | |
| Across 6 months[d] | | −0.10 (−0.29 to 0.10), 0.50 |
| Across 24 months[c] | 0.94 (0.49–1.81), 0.85 | |

[a]Adjusted for baseline detectable pVL and age.

[b]Adjusted for baseline pVL ($\log_{10}$ continuous) and age.

[c]Adjusted for baseline detectable pVL, any concurrent RTI, and age.

[d]Adjusted by baseline continuous pVL ($\log_{10}$ continuous), any concurrent RTI, and age.

AOR, adjusted odds ratio; ART, antiretroviral therapy; CI, confidence interval;; C-IUD, copper intrauterine device; LNG-IUS, levonorgestrel intrauterine system; pVL, plasma viral load; RTI, reproductive tract infection

For an objective measure of bleeding, we compared hemoglobin change between arms in AT analysis to present relative change concomitant with IUC use. Generally, change in hemoglobin from baseline was significantly higher among LNG-IUS users compared with C-IUD users (Table 6). Changes were markedly higher at the 6-month visit than at the 24-month visit for non-ART women, compared with steady increases in ART users using the LNG-IUS.

## IUC expulsion and discontinuation

IUC continuation was 78% (155/199) across all participants during the cohort period; the overall IUC expulsion rate was 2.8/100 p-y, 95% CI 1.5–5.5. Expulsion (5.6/100 p-y, 95% CI 2.8–11.1 for C-IUD versus 0.6/100 p-y, 95% CI 0.10–4.1 for LNG-IUS, $p$ = 0.03) and all-cause discontinuation rates (35% for C-IUD versus 6% for LNG-IUS, $p \leq$ 0.001; Fig 3) were significantly higher among women using the C-IUD.

Elective discontinuation hazards were significantly higher for C-IUD (aHR = 8.61, 95% CI 3.03–24.4, $p$ < 0.001) users. Overall IUC discontinuation did not differ significantly by ART status (aHR = 1.12, 95% CI 0.11–11.40, $p$ = 0.92), though non-ART women tended to request discontinuation earlier than ART users (S1 Fig).

## Discussion

We found no significant difference in proportions of women with detectable gVL between LNG-IUS and C-IUD arms and no change in detectable gVL prior to insertion compared across 3 and 6 months following insertion of either IUC. Similarly, there was no significant difference in proportions of women with detectable pVL (for ART users) or mean change in $\log_{10}$ pVL (for non-ART users) between IUC arms. We also demonstrated that the LNG-IUS had

**Table 5. Adverse events compared between women living with HIV using the LNG-IUS or C-IUD, by ART status ($n = 199$).**

| Adverse events | All participants ($n = 199$) | | | ART-using women ($n = 132$) | | | Non-ART women ($n = 67$) | | |
|---|---|---|---|---|---|---|---|---|---|
| | LNG-IUS users with ≥1 event; $n$ (%) | C-IUD users with ≥1 event; $n$ (%) | $p$-value | LNG-IUS users with ≥1 event; $n$ (%) | C-IUD users with ≥1 event; $n$ (%) | $p$-value | LNG-IUS users with ≥1 event; $n$ (%) | C-IUD users with ≥1 event; $n$ (%) | $p$-value |
| **Overall events** | 99 (98.0) | 96 (98.0) | 1.00 | 63 (96.9) | 65 (97.0) | 1.00 | 36 (100) | 31 (100) | 1.00 |
| **Events potentially related to intrauterine contraception** | | | | | | | | | |
| **Bleeding conditions** | | | | | | | | | |
| Amenorrhea | 49 (48.5) | 11 (11.2) | <0.001 | 32 (49.2) | 8 (11.9) | <0.001 | 17 (47.2) | 3 (9.7) | 0.001 |
| Menorrhagia | 5 (5.0) | 28 (28.6) | <0.001 | 3 (4.6) | 17 (25.4) | 0.001 | 2 (5.6) | 11 (35.5) | 0.004 |
| Intermenstrual bleeding | 23 (22.8) | 22 (22.4) | 1.00 | 13 (20.0) | 17 (25.4) | 0.54 | 10 (27.8) | 5 (16.1) | 0.38 |
| Irregular/heavy bleeding | 8 (7.9) | 13 (13.3) | 0.25 | 4 (6.2) | 7 (10.4) | 0.53 | 4 (11.1) | 6 (19.4) | 0.49 |
| Menstrual disorder | 7 (6.9) | 5 (5.1) | 0.77 | 7 (10.8) | 5 (7.5) | 0.56 | 0 (0) | 0 (0) | – |
| **Other gynecologic conditions: pain** | | | | | | | | | |
| Pelvic pain | 29 (28.7) | 30 (30.6) | 0.88 | 12 (18.5) | 15 (22.4) | 0.67 | 17 (47.2) | 15 (48.4) | 1.00 |
| Dysmenorrhea | 6 (5.9) | 15 (15.3) | 0.04 | 3 (4.6) | 8 (11.9) | 0.21 | 3 (8.3) | 7 (22.6) | 0.17 |
| Enlarged abdomen | 6 (5.9) | 7 (7.1) | 0.78 | 3 (4.6) | 2 (3.0) | 0.68 | 3 (8.3) | 5 (16.1) | 0.46 |
| Breast pain | 3 (3.0) | 7 (7.1) | 0.21 | 0 (0) | 1 (1.5) | 1.00 | 3 (8.3) | 6 (19.4) | 0.28 |
| Lower back pain | 5 (5.0) | 2 (2.0) | 0.45 | 4 (6.2) | 2 (3.0) | 0.44 | 1 (2.8) | 0 (0) | 1.00 |
| **Other gynecologic conditions: symptoms of infection/inflammation** | | | | | | | | | |
| Leukorrhea | 5 (5.0) | 10 (10.2) | 0.19 | 2 (3.1) | 4 (6.0) | 0.68 | 3 (8.3) | 6 (19.4) | 0.28 |
| Cervicitis | 2 (2.0) | 1 (1.0) | 1.00 | 2 (3.1) | 1 (1.5) | 0.62 | 0 (0) | 0 (0) | – |
| **Systemic conditions** | | | | | | | | | |
| Weight gain | 11 (10.9) | 13 (13.3) | 0.67 | 5 (7.7) | 5 (7.5) | 1.00 | 6 (16.7) | 8 (25.8) | 0.38 |
| Weight loss | 2 (2.0) | 1 (1.0) | 1.00 | 0 (0) | 0 (0) | – | 2 (5.6) | 1 (3.2) | 1.00 |
| Headache | 9 (8.9) | 7 (7.1) | 0.80 | 7 (10.8) | 3 (4.5) | 0.20 | 2 (5.6) | 4 (12.9) | 0.40 |
| Anemia | 3 (3.0) | 10 (10.2) | 0.05 | 2 (3.1) | 8 (11.9) | 0.10 | 1 (2.8) | 2 (6.5) | 0.59 |
| Nausea | 7 (6.9) | 6 (6.1) | 1.00 | 2 (3.1) | 5 (7.5) | 0.44 | 5 (13.9) | 1 (3.2) | 0.21 |
| Acne | 0 (0) | 4 (4.1) | 0.06 | 0 (0) | 4 (6.0) | 0.12 | 0 (0) | 0 (0) | – |

ART, antiretroviral therapy; C-IUD, copper intrauterine device; LNG-IUS, levonorgestrel intrauterine system

significantly higher continuation during the study (96% for LNG-IUS versus 59% for C-IUD), reflecting greater acceptability of the LNG-IUS than the C-IUD. There were few cases of PID, and the IUC had a high overall continuation rate, which is notable among a population with high background rates of curable RTIs. We achieved our study aim to measure IUC safety with regard to HIV disease and AEs, as well as relative acceptability through continuation rates, although the sample size in the non-ART group was smaller than originally planned.

We selected gVL as a proxy measure for transmission risk given association between gVL and sexual transmission to male partners in discordant couples [12,38]. There are limited data regarding quantity at which gVL substantively increases transmission risk, though 1 publication suggests transmission is unlikely below 400 copies/mL in ART-using populations [39]. Our safety findings for odds of having detectable gVL did not change with exposure to either IUC or between study arms across 6 and 24 months even after adjusting for ART use, RTI, and pVL, and in sensitivity analyses adjusted for disproportionate IUC discontinuation between arms. Chinula and colleagues found reduced proportions of women with detectable gVL following LNG implant use at 6 months for tear flow strip samples (17.7% and 3.2%, representing

**Table 6. Mean relative change in hemoglobin level between women living with HIV using the LNG-IUS or C-IUD across 6 and 24 months, by ART use (*n* = 186).**

| | All participants (*n* = 186) | ART-using women (*n* = 125) | Pre-ART women (*n* = 61) |
|---|---|---|---|
| **Hemoglobin by study visit** | LNG-IUS versus C-IUD (95% CI) *p*-value | | |
| **As-treated analysis[a]** | | | |
| Across 6 months | 0.57 (0.24–0.90) <0.001 | 0.55 (0.11–0.99) 0.014 | 0.63 (0.20–1.06) 0.004 |
| Across 24 months | 0.71 (0.47–0.95) <0.001 | 0.77 (0.45–1.08) <0.001 | 0.57 (0.25–0.89) <0.001 |
| **As-treated analysis[b]** | | | |
| Across 6 months | 0.58 (0.25–0.91) <0.001 | 0.57 (0.13–1.02) 0.012 | 0.64 (0·21–1.06) 0.004 |
| Across 24 months | 0.71 (0.47–0.95) <0.001 | 0.78 (0.47–1.10) <0.001 | 0.57 (0.25–0.89) <0.001 |

[a]Adjusted for baseline hemoglobin in generalized linear model.

[b]Adjusted for baseline hemoglobin, age, pVL, and (pooled) ART status in generalized linear model.

ART, antiretroviral therapy; CI, confidence interval; C-IUD, copper intrauterine device; LNG-IUS, levonorgestrel intrauterine system; pVL, plasma viral load

follicular and luteal phase measurements prior to method initiation at baseline, versus 6.1% at 180 days; adjusted risk ratio [aRR] = 0.40 [95% CI 0.18–0.85], adjusted for baseline pVL and CD4 count), compared with no significant change in women using DMPA (12.1% and 10.7% at baseline versus 10.3% at 180 days, aRR = 1.37 [95% CI 0.81–2.33]) among WLHIV reporting

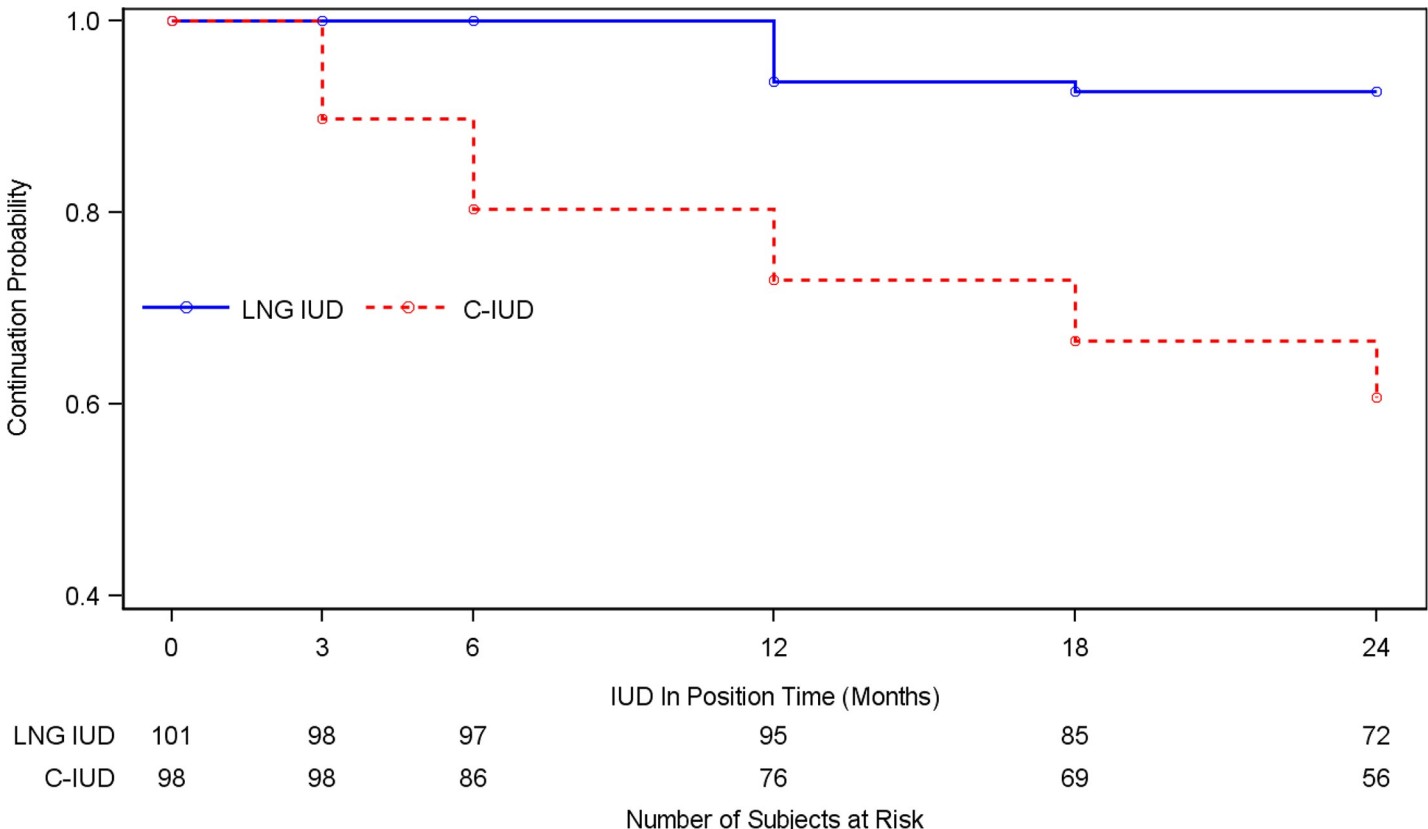

**Fig 3. Kaplan-Meier curve comparing IUC continuation rates between LNG-IUS and C-IUD users overall among women living with HIV in Cape Town, South Africa (*n* = 199).** C-IUD, copper intrauterine device; IUC, intrauterine contraceptive; IUD, intrauterine device; LNG IUS, levonorgestrel intrauterine system.

ART use in Malawi [40]. For the same cohort followed up to a maximum of 33 months, Kourtis and colleagues found that risk of detectable gVL did not differ significantly over time after initiation of either method (aRR, 0.97; 95% CI 0.84–1.13, per increase of 6 months of follow-up) or between the DMPA study arm compared with the LNG implant study arm (aRR, 1.92; 95% CI 0.97–3.79) [37]. In this extended cohort period, detectable gVL also did not significantly differ before method initiation compared with follow-up visits ($p$ = 0.47). In a prospective cohort of WLHIV initiating ART that measured detectable gVL, defined as >400 copies/mL from endocervical swab, 30.3% of women reporting ART use had detectable gVL at 1 or more visits; and DMPA (OR 0.96, 95% CI 0.44–2.13) exposure was not associated with detectable gVL [34]. In a Burkina Faso cohort of women with high background rates of RTI, 45% (77/170) of women had detectable gVL at 1 or more visits despite ART use with undetectable pVL [22]. In this cohort, neither reported oral contraceptive (aOR = 1.57, 95% CI 0.75–3.27) nor DMPA use (aOR = 1.32, 95% CI 0.42–4.16) was associated with detectable gVL in analysis adjusted for pVL. Our findings bear distinction from a mucosal immunology standpoint, as the LNG-IUS has markedly higher LNG concentrations in genital tract mucosa, reflecting its largely localized effect, in contrast to the LNG implant and DMPA [41]. When compared with the C-IUD, LNG does not appear to drive shedding via gVL; neither IUC appears to increase gVL through hypothesized inflammatory response from foreign body effect.

There was no significant detectable effect of levonorgestrel from the LNG-IUS on pVL measures between study arms within each ART group, consistent with previous studies [10,11]. Additionally, we found that more than 15% of ART users had pVL > 1,000 copies/mL at any visit after enrollment, consistent with trends attributed to ART nonadherence in Cape Town and the association of detectable gVL with detectable pVL in other studies assessing impact of contraception on genital tract shedding [34,37,40,42]. This finding was associated with increased proportions of women with detectable gVL, irrespective of IUC type and potentially augmented by incident RTIs. Given frequent detectable pVL levels that reflect lack of clinical viral suppression with "real-world ART use" and thus increased probability of transmission among our participants, our findings that neither IUC increased detectable gVL to further increase transmission risk is clinically important. Further, these pVL trends emphasize the need to ensure WLHIV are counseled regarding condom use to prevent transmission to partners when their HIV is not fully suppressed.

We found low numbers of related SAEs, particularly PID, despite relatively high RTI prevalence within our cohort. There were 3 PID cases, occurring in both arms and at levels similar to those reported by studies with comparator groups in some contexts [8,43] but higher than in others [17,44,45]. The cohorts with lower rates were European WLHIV comprising mainly experienced ART users with no reported PID cases [14], a postpartum Zambia cohort recruited prior to Option B+ ART coverage [44], and a Ugandan cohort [45]. For the Zambian study, we believe recruiting postpartum women with no history of PID reduced PID risk because of possible reduced sexual activity in the postpartum period and thus lower exposure to pathogens, evidenced by only 1 case of PID within that cohort of 296 women contributing 642 p-y (0.16/100 p-y) of follow-up [44]. The Ugandan cohort reported 5 cases (2 LNG-IUS and 3 C-IUD) of PID among 672 participants (incidence rate ratio = 0.7, 95% CI 0.06–6.04) with all cases diagnosed and managed by the study site [45]. Our 3 cases include 2 that were diagnosed and managed at other clinical sites, and it is possible that IUC use may have predisposed the external provider to PID diagnosis, as noted in other settings [46,47]. RTIs were measured, detected, and treated at screening (prevalence data); continued to occur through the cohort period; and were largely asymptomatic with few findings on exam [48]. Given the high prevalence of asymptomatic infections, RTI testing using point-of-care tests rather than syndromic management (the standard of care) should be added within IUC insertion

guidelines to reduce risk of PID specifically related to IUC use and thus address some provider concerns that likely hamper offering IUCs to WLHIV [20,49]. There were 2 ectopic pregnancies, both occurring with the C-IUD, a propensity noted previously and possibly magnified in this cohort because of high RTI prevalence with tubal scarring [50]. The overall pregnancy rate was higher than expected based on some studies that found pregnancy rates between 0.06 and 1.7/100 p-y [45,50,51] but lower than the 3-year cumulative rate for C-IUD users noted in a large multicenter cohort of 2.8/100 p-y (95% CI 1.3–6.0) [43]. We attribute some of the pregnancies and expulsions to IUC malposition, which may have been clinically unrecognizable until the IUC was visible at the cervical os; Bahamondes and colleagues noted similar rates among C-IUD users, which they attributed to issues related to variability in inserter design [43]. C-IUD users had higher expulsion rates, similar to levels reported in other studies [45,52] but in contrast to that noted by Sanders and colleagues in a US cohort, where expulsion rates were 9% at 12 months and significantly higher among LNG-IUS users [51]. We noted partial expulsions during scheduled study visits, particularly for the C-IUD whose narrow inserter may have resulted in clinicians hesitating to move the inserter all the way to the fundus and then back by 1 cm before advancing the IUD into the endometrial cavity because of perforation concerns. AEs potentially related to IUC use were significantly higher for C-IUD users with the exception of amenorrhea, which was higher for LNG-IUS users, and pelvic pain, which was similar between arms. Amenorrhea was not a strong motivator for discontinuation and may have been under-reported as an AE based on individual participant perception. By contrast, menorrhagia was frequently cited as reason for discontinuation based on AE distribution and stated reasons for C-IUD removal.

WLHIV allocated to the LNG-IUS generally had significant and steady increases in hemoglobin [53]. This increase was more dramatic among non-ART women in the first 6 months than in the interval between 6 and 24 months and may reflect interaction with untreated disease, as many women enrolled in the non-ART group later reported ART use, evidenced by increase in proportions with undetectable pVL by 18 months.

In this trial, the primary and secondary outcomes focused on ensuring IUC safety for WLHIV before and during ART use; however, as IUCs are meant to be LARC methods, acceptability is critical in determining actual use. In this context, the significantly higher LNG-IUS continuation rate observed here is notable. Kakaire and colleagues found similar continuation rates between the C-IUD and LNG-IUS, both exceeding 90%, among WLHIV in Uganda across 12 months [43]. We found much higher LNG-IUS continuation, and among women requesting IUC removal, mean time to discontinuation for both IUCs was less than 1 year, which may reflect differing contextual norms around tolerance of contraceptive side effects or perceived contraceptive availability [43,52].

Our IUC study is among few whose follow-up duration exceeds 12 months, and the high LNG-IUS continuation rate documented here should contribute to expanding access to this LARC method, permitting women broader choice across the reproductive period. Although the LNG-IUS is more expensive than the C-IUD, the significantly higher continuation rate may lead to increased cost-effectiveness relative to other LARC methods that have higher discontinuation rates and thus provide a strong rationale for including the LNG-IUS as a contraceptive option in low- and middle-income countries. This finding is important, as newer versions of the LNG-IUS, bio-identical to the predominant marketed product and approved by regulatory bodies, have been introduced recently with a substantially lower unit cost [54]. These products may potentially make widespread access to LNG-IUS contraception possible, similar to market changes that enabled increased global access to the contraceptive implant [55].

The strengths of this study include the masking of participants and outcome assessors, the randomized design, the inclusion of a nonhormonal method control, the inclusion of both ART and non-ART users, and use of menstrual cup specimens for gVL measurement [30]. However, several limitations are important to note. Our sample size was small for the non-ART group, which was difficult to enroll in light of changing ART guidelines. We also note that among non-ART women screened, many were not eligible because of low CD4 lymphocyte counts constituting ART eligibility by guidelines in effect at study start in 2013. We hypothesize that some non-ART WLHIV preferred not to disclose their status and hence avoided any clinical setting related to HIV care, including the clinic where this study's clinical activities were conducted. Additionally, a further 17 women were not eligible for their 24-month visit at study closure, resulting in truncated 24-month visit outcome data. We conducted telephone follow-up of these women for the IUC continuation measure when they would have been eligible for the 24-month visit and reached 11 of 17 women, but VL data were missing. Last, the VL assay used by NHLS changed during the study period with VL measures conducted on 2 separate assays for most participants. These assays are both validated but have different LoDs and may have resulted in classification bias with differing precision around VL values.

These data suggest that, compared with the C-IUD, the LNG-IUS does not increase gVL or pVL and has low levels of contraceptive failure and associated PID in WLHIV with high background rates of RTIs. The LNG-IUS had significantly lower overall and elective discontinuation and expulsion rates, suggesting it is a highly acceptable method for WLHIV. These findings may be used to strengthen policies related to IUCs for WLHIV as well as to educate HIV and family planning service providers, and WLHIV themselves, regarding IUCs as a safe contraceptive option.

## Supporting information

**S1 CONSORT checklist. CONSORT 2010 checklist of information to include when reporting a randomized trial.**
(DOC)

**S1 Table. Odds of detectable genital tract viral load comparing women using the LNG-IUS with those using the C-IUD with linear regression weighted for differential intrauterine contraceptive discontinuation rates, stratified by ART status, among women living with HIV in Cape Town, South Africa.** ART, antiretroviral therapy; C-IUD, copper T-380 intrauterine device; LNG-IUS, levonorgestrel intrauterine system
(DOCX)

**S2 Table. Odds of detectable genital tract viral load comparing women using the LNG-IUS with those using the C-IUD with linear regression adjusted by visit month, stratified by ART status, among women living with HIV in Cape Town, South Africa.** ART, antiretroviral therapy; C-IUD, copper T-380 intrauterine device; LNG-IUS, levonorgestrel intrauterine system
(DOCX)

**S3 Table. Odds of detectable genital tract viral load comparing women using the LNG-IUS with those using the C-IUD with linear regression using working independent correlation structure, stratified by ART status, among women living with HIV in Cape Town, South Africa.** ART, antiretroviral therapy; C-IUD, copper T-380 intrauterine device; LNG-IUS, levonorgestrel intrauterine system
(DOCX)

**S4 Table. Odds of detectable pVL for women using ART or difference in mean change of log10 pVL among women not using ART at enrollment, comparing women using the LNG-IUS with those using the C-IUD, with linear regression weighted for differential intrauterine contraceptive discontinuation rates, among women living with HIV in Cape Town, South Africa.** ART, antiretroviral therapy; C-IUD, copper T-380 intrauterine device; LNG-IUS, levonorgestrel intrauterine system; pVL, plasma viral load
(DOCX)

**S5 Table. Odds of detectable pVL for women using ART or difference in mean change of log10 pVL among women not using ART at enrolment, comparing women using the LNG-IUS with those using the C-IUD, with linear regression adjusted by visit month, among women living with HIV in Cape Town, South Africa.** ART, antiretroviral therapy; C-IUD, copper T-380 intrauterine device; LNG-IUS, levonorgestrel intrauterine system; pVL, plasma viral load
(DOCX)

**S6 Table. Odds of detectable pLV for women using ART or difference in mean change of log10 pVL among women not using ART at enrolment, comparing women using the LNG-IUS with those using the C-IUD, with linear regression using working independent correlation structure, among women living with HIV in Cape Town, South Africa.** ART, antiretroviral therapy; C-IUD, copper T-380 intrauterine device; LNG-IUS, levonorgestrel intrauterine system; pVL, plasma viral load
(DOCX)

**S1 Fig. Kaplan-Meier continuation rate estimates for women using the LNG-IUS compared with women using the C-IUD among women living with HIV in Cape Town, South Africa, by antiretroviral therapy status.** C-IUD, copper T-380 intrauterine device; LNG-IUS, levonorgestrel intrauterine system
(TIF)

**S1 Text. Protocol: Comparison of Two IUDs among Cape Town HIV-positive Women: A Randomized Controlled Trial Assessing Safety of Registered Products in South Africa.** FHI 360 Study 10369, Version 10.0.
(PDF)

**S2 Text. Comparison of Two IUDs among Cape Town HIV-positive Women: A Randomized Controlled Trial Assessing Safety of Registered Products in South Africa.** Statistical Analysis Plan, Version 9.0.
(PDF)

## Acknowledgments

We thank our participants for their time and trust. We thank our study staff, the staff of the Gugulethu Green Clinic, CIDER staff, and microbiology and NHLS laboratory colleagues for their assistance and efforts. We thank Jaim Jou Lai for analysis assistance and Kerry Aradhya and Anna Cook for formatting assistance and for collegial review from Douglas Taylor, Charles Morrison, and Timothy Mastro.

## Author Contributions

**Conceptualization:** Catherine S. Todd, Landon Myer.

**Data curation:** Heidi E. Jones.

**Formal analysis:** Heidi E. Jones, Donald R. Hoover, Pai-Lien Chen.

**Funding acquisition:** Catherine S. Todd, Landon Myer.

**Investigation:** Landon Myer.

**Methodology:** Catherine S. Todd, Heidi E. Jones, Donald R. Hoover, Landon Myer.

**Project administration:** Nontokozo Langwenya.

**Resources:** Gregory Petro.

**Software:** Heidi E. Jones.

**Supervision:** Nontokozo Langwenya, Gregory Petro, Landon Myer.

**Validation:** Nontokozo Langwenya, Pai-Lien Chen.

**Writing – original draft:** Catherine S. Todd.

**Writing – review & editing:** Heidi E. Jones, Nontokozo Langwenya, Donald R. Hoover, Pai-Lien Chen, Gregory Petro, Landon Myer.

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
