## [Decision Letter · Decision Letter 0]

15 Jan 2020

Dear Dr. Todd,

Thank you very much for submitting your manuscript "Comparison of safety and continuation of the levonorgestrel intrauterine system versus the copper intrauterine device among women living with HIV in South Africa: a randomized controlled trial" (PMEDICINE-D-19-04204) for consideration at PLOS Medicine. 

Your paper was evaluated by an academic editor with relevant expertise and sent to independent reviewers, including a statistical reviewer. The reviews are appended at the bottom of this email and any accompanying reviewer attachments can be seen via the link below:

[LINK]

In light of these reviews, we will not be able to accept the manuscript for publication in the journal in its current form, but we would like to invite you to submit a revised version that fully addresses the reviewers' and editors' comments. You will appreciate that we cannot make a decision about publication until we have seen the revised manuscript and your response, and we expect to seek re-review by one or more of the reviewers. 

We hope to receive your revised manuscript by Feb 05 2020 11:59PM. Please email us (plosmedicine@plos.org) if you have any questions or concerns.

Please let me know if you have any questions. Otherwise, we look forward to receiving your revised manuscript in due course. 

Sincerely,

Richard Turner, PhD

rturner@plos.org

Please finalize the arrangements for data deposition.

If devices used in the study were donated by commercial entities, please state this in the financial disclosure (metadata). 

Please add the start and end dates of participant recruitment to your abstract, and identify the primary endpoint early in the "methods and findings" subsection.

In the abstract, please add a few words to clarify "related SAEs".

Please quote 1-2 further study limitations in the abstract. 

In the "conclusions" subsection of your abstract, please adapt the wording of the first sentence to use the past tense, e.g., "In this study, we found that ... did not increase ... and had ...".

In the abstract and throughout the paper, please add p values alongside 95% CI where available. 

After your abstract, please add a new and accessible "author summary" section in non-identical prose. You may find it helpful to consult one or two recent research papers in PLOS Medicine to get a sense of the preferred style. 

Please remove the final sentence of the introduction; you may wish to include this element of discussion later in the paper. 

Please remove the "role of the funding source" information from the text (this information will be included in the article metadata in the event of publication). 

Throughout the ms, please quote p values as p<0.001 or exact values. 

Please remove trade marks, throughout the article. 

Please add additional access details to reference 22, as needed. 

Please include a completed CONSORT checklist as a supplementary document, referred to in the methods section of your main text. In the checklist, individual items should be referred to by section (e.g., "Methods") and paragraph number rather than by page or line numbers, as the latter generally change in the event of publication. 

Comments from the reviewers:

*** Reviewer #1: 

The authors present the findings for a randomised controlled study comparing viral load in two different contraceptive strategies in women with HIV in South Africa: LNG-IUS and C-IUD in stratified by both ART and non-ART groups. The findings show that there was no difference in outcomes in HIV viral load outcomes between interventions in both ART and non-ART groups. The primary limitation was, as the authors mentioned, not achieving the target sample size in the non-ART group due to guidelines changing during the trial. The authors did a nice job following CONSORT reporting guidelines and appreciate that the authors were transparent in their decision making regarding the rational for changing the primary outcome from pVL to gVL due to a guideline recommendation change (with pVL retained as a secondary outcome). Given the lack of trials in this area, with low quality observational studies to date, this study should merit publication and be informative to many in this field. I have a few comments that could be considered. Given the stratified analyses by non-ART/ART groups, generalised linear models are appropriate model of choice here. 

1) Abstract: it would helpful to report the allocation ratio here as well

2) Introduction: From the introduction, it appears from the literature of other small observational there is some uncertainty in either LNG-IUS compared to C-IUD. I just want to understand the hypotheses better, that given some uncertainty of previous studies described in the introduction. Are the authors saying that null is no difference in primary outcomes with the alternative being "there is a difference between primary outcomes" - typical superiority design OR Are they suggesting that the null is treatments differ and the alternative hypothesis is the treatments are same within a tolerable margin? - non-inferiority design. It would good if the authors can clarify this. 

3) Introduction: Last sentence - this should go in the discussion as a limitation instead of the introduction

4) Methods - statistical analysis: As the authors will probably appreciate, at-treated analysis is aimed to estimate the effect of the treatment "as delivered" or "as recieved" opposed to ITT "as assigned" to account for treatment non-adherence. Non-ITT or as-treated analysis usually presented as secondary analysis but in this paper it is presented the other way around. I can understand why this was done given the authors captured adherence parameters but i think the authors should give some rationalisation towards this decision in the analysis section to help understanding. 

5) Methods -statistical analysis: Sentence beginning with "adjusted models for both ART groups..." referring to pre-specified confounders - can you refer to this in the pre-specified analysis plan. 

6) Methods - statistical analysis: Statement on participants with "undetectable" or lower than LoD results being assigned values...this needs some justification or reference associated with this decision

7) Methods - statistical analysis: On methods regarding Cox models for discontinuation - please indicate if these adjustment for covariates were pre-specified and proportional hazards assumptions were checked (i.e. Schoenfeld residuals)

8) Results - CONSORT flow diagram: indicate in flow diagram during follow-up boxes in each arm that the numbers included for the primary analyses were analysed in as-treated or non-ITT analysis either as * or footnote

9) Results - statement on Page 14 regarding clinical activity closures - It's unclear what this means - is this study closure and the participants were unmasked, therefore they could not be eligible? Is this the correct interpretation?

Overall, this is a nicely presented trial and should be informative for clinicians and health care policy-makers. 

*** Reviewer #2: 

This is a clear, well-written manuscript describing important results from an RCT evaluating impacts of IUD (LN vs copper) on acceptability, plasma HIV RNA, and genital HIV RNA shedding. The manuscript would be strengthened by additional attention to the reason gVL was chosen as the primary outcome, the clinical significance of detectable gVL, and the significance of the lack of a difference in gVL for the question of HIV transmission among women using IUD. Additional comments below:

--An inclusion criterion was to be screened with cervical cytology within the last year. Given the poor prevalence of cervical ca screening in SA, how does this impact the generalizability of the findings?

--Another inclusion criterion was interest in IUC use for contraception. How unique is this group?

-- Please provide additional information about why detectable gVL was chosen as an outcome and what is known about what absolute levels are required for efficient transmission. There are a lot of data re. genital HIV shedding in the presence of plasma HIV-RNA suppression - and yet the data re plasma undetectable = uninfectious are compelling and now accepted by most as "trumping" information re. genital VL. What are the data that suggest that gVL matters for this question? And what were the cut-offs and what data inform that? And for power, what is the meaning of a 19% difference in detectable gVL? And why is the assumed pooled baseline proportion or women with detectable gVL estimated to be 42.8%? Overall, given that gVL is the primary outcome, it would be helpful to say a bit more about this measure and the existing data that should inform interpretation of these data in the intro/discussion.

- Methods state, "Due to clinical activity closure, 19 women (17 ART users, two non-ART users) were not eligible for the 24-month visit." Please clarify what this means.

- An important point for methods and discussion - were women exited when they discontinued IUD? If so, that would likely alter (increase) continuation rates given 2dary benefits of being in a clinical trial in RSA.

- An important point that is missing from the current version of the discussion is that 40% of women had a curable RTI which may not have been picked up via standard of care syndromic screening. Contraceptive programs should be supported by MOH to implement testing protocols.

- Table 7 the back-slash (/) before p-value is odd looking in table. This can be managed in journal formatting.

- Consort Figure 1 - The reason for 21 of those excluded from the pre-ART arm is "other". Could you describe since this is 10% of those screened?

*** Reviewer #3: 

This is a well conducted, double blinded, randomized trial of the LNG versus vs copper IUD in HIV-infected women in South Africa. It provides important evidence on the acceptability, efficacy, and safety of these two underutilized contraceptive methods in a population that could greatly benefit from more options. The manuscript follows CONSORT guidelines for reporting. The analysis adheres to a priori study questions and follows the plan outlined in protocol version 10, which was provided alongside the study materials. The conclusions drawn are reasonable and the trial strengths and weaknesses are adequately addressed.

* It is unclear in Figure 1 whether the 6 women who declined, were ineligible, or who could not have the IUD inserted are post-randomization. If so, these women should be included in the ITT analysis.

* Table 2 is dense and I found it difficult to interpret. The reader is left to calculate the "deltas" on his/her own. Would it not work better to split it into a figure for the visit trends and then report the 6- and 24-month deltas in a separate table? I'm not really sure, but Table 2 seems to bury your primary results. Also, it is confusing how the "% (n)" designation in table 2 differs from table 1, which uses "n (%)."

* While there was no difference observed between the two methods with respect to genital HIV shedding, it might be informative to the reader to provide comparative data from HIV infected women on other methods. I found myself wondering whilst reading the discussion whether I should be concerned that 10-30% of women on ART had detectable virus in their vaginal samples. How does this compare to other contraceptive methods?

* The frequency of PID in this study, while low, is higher than that reported from analogous trials in Zambia (not referenced, but probably should be) and Uganda (ref #26). It might be helpful to report this outcome as a rate (events/person-years of follow-up) to allow comparison with other research. 

* The pregnancy rate also seems higher than expected. The results might benefit from conversion to rates and the discussion from a few point estimates from other sources for comparison (even if only available in HIV- women). In fact, for each of the important events: continuation, expulsion, pregnancy, PID - it would be helpful to have these reported as rates so that the reader could compare to other papers.

* I don't understand why 19 women were "not eligible" for their final visit. 

* There's a statement on pp 25-26 that says: "Among women discontinuing their allocated IUC, median time to elective discontinuation was significantly shorter for C-IUD users (7.2 months for C-IUD vs 11.2 months for LNG-IUS; Fig 2)." It doesn't seem appropriate (or necessary) to report this rate among women who discontinued their method, since it would no longer be a randomized comparison. One way to report this and maintain the randomization would be to right censor those who continue at 24 months (like Figure 2 does) and report median time on contraception. If you do decide it is important to include this calculation, I would advise not referring to Figure 2, since that KM appears to include all women. 

* CONSORT calls for the discussion to address sources of "potential bias, imprecision, and, if relevant, multiplicity of analyses." The imprecision issue seems particularly relevant here. For instance, the proportion of women on ART with detectable gVL was increased nearly 3-fold between randomization and 24 months in the cIUD group, compared to very little change in the LNG-IUG group. With a larger sample, would we be worried about this? 

* Finally, while the trial analysis follows that which is outlined in the protocol, it is unclear why the authors opted for an adjusted analysis as primary. The randomization procedures described were convincing and I found myself most interested in the unadjusted ITT analysis. Perhaps the pooling approach mandates this, but I think most clinical readers will search for unadjusted head-to-head comparisons.

***

[LINK]

---

## [Decision Letter · Decision Letter 1]

18 Mar 2020

Dear Dr. Todd,

Thank you very much for re-submitting your manuscript "Comparison of safety and continuation of the levonorgestrel intrauterine system versus the copper intrauterine device among women living with HIV in South Africa: a randomized controlled trial" (PMEDICINE-D-19-04204R1) for consideration at PLOS Medicine.

I have discussed the paper with our academic editor and it was also seen again by one reviewer. I am pleased to tell you that, provided the remaining editorial and production issues are dealt with, we expect to be able to accept the paper for publication in the journal.

[LINK]

Please let me know if you have any questions. Otherwise, we look forward to receiving the revised manuscript shortly. 

Kind regards,

Richard Turner, PhD

rturner@plos.org

Requests from Editors:

Please finalize the arrangements for data deposition. 

As a general observation, we suggest mentioning the non-inferiority design (line 154) in the abstract. In places (e.g. line 65, "did not increase") the wording might suggest to some readers that a superiority trial was done, and we suggest some judicious rewording (e.g., "there was no significant difference"). 

On this same point, around line 250 the wording seems to imply that your aim was to "detect a difference", and we suggest amending this to "... gave 80% power to exclude a difference of 27.7% or more", for example. Are you able to state a non-inferiority margin for the primary endpoint assessment in your methods section?

We suggest amending the title to "Safety and continued use of the levonorgestrel intrauterine system as compared with the copper intrauterine device among women living with HIV in South Africa: a randomized controlled trial" or similar. 

Please trim the discussion of limitations in your abstract; for example, the comment about change of assay could be compressed to "... and a change in the assay used for viral load testing during the study" or similar. 

Where you make claims such as "the first" (e.g., at line 98) please add "to our knowledge" or similar. 

Please remove the "TM" symbol at line 199 and any other instances (including in the metadata).

Please refer to the attachments (protocol, statistical analysis plan and CONSORT checklist) at suitable points in the methods section of your main text. 

We suggest harmonizing the study start and end dates quoted in your abstract and results section. 

Comments from Reviewers:

*** Reviewer #1: 

The authors have addressed all the revisions and made the necessary changes to the manuscript. No additional comments to address

***

[LINK]

---

## [Editor Report · Decision Letter 2]

13 Apr 2020

Dear Dr. Todd, 

On behalf of my colleagues and the academic editor, Dr. Lynne Meryl Mofenson, I am delighted to inform you that your manuscript entitled "Safety and continued use of the levonorgestrel intrauterine system as compared with the copper intrauterine device among women living with HIV in South Africa: a randomized controlled trial" (PMEDICINE-D-19-04204R2) has been accepted for publication in PLOS Medicine. 

PRODUCTION PROCESS

PRESS

PROFILE INFORMATION

Thank you again for submitting the manuscript to PLOS Medicine. We look forward to publishing it. 

Best wishes, 

Richard Turner, PhD

Senior Editor 

PLOS Medicine

plosmedicine.org